# Lagochilascariasis: A Neglected Zoonosis in the Brazilian Amazon Biome and the Role of Wildlife in Its Epidemiological Chain Amidst Anthropization

**DOI:** 10.3390/tropicalmed10070177

**Published:** 2025-06-21

**Authors:** Felipe Masiero Salvarani, Karoline Petrini Pinheiro da Cruz, Flavio Roberto Chaves da Silva, Cíntia Daudt

**Affiliations:** 1Laboratório de Medicina Veterinária Preventiva, Instituto de Medicina Veterinária, Universidade Federal do Pará, Castanhal 68740-970, PA, Brazil; karolinepetrini@gmail.com; 2Laboratório de Virologia Geral e Parasitologia, Centro de Ciências Biológicas e da Natureza, Universidade Federal do Acre, Rio Branco 69920-900, AC, Brazil; flavio.silva@ufac.br (F.R.C.d.S.); cintia.daudt@ufac.br (C.D.)

**Keywords:** One Health, wildlife reservoirs, helminthiasis, zoonotic transmission

## Abstract

Lagochilascariasis is a neglected zoonotic helminthiasis, caused by *Lagochilascaris minor*, characterized by a complex and not well understood transmission cycle. This parasitic disease is endemic to Latin America, particularly Brazil, and is associated with rural and forested areas, where humans may serve as accidental hosts. The southeastern region of Pará state reports the highest number of cases, highlighting its epidemiological significance. Wildlife species, especially carnivores and rodents, play crucial roles as definitive and intermediate hosts, respectively. Although lagochilascariasis can lead to severe clinical manifestations, including chronic soft tissue infections and potential central nervous system involvement, it is likely underdiagnosed due to its similarity to fungal and bacterial diseases. The anthropization of the Amazon Biome, through deforestation and habitat fragmentation, coupled with increased human–wildlife interactions, may be influencing the epidemiology of this parasitosis. This review aims to summarize current knowledge of *L. minor* transmission routes, the role of wildlife in maintaining its cycle, and the impact of environmental changes on disease patterns. Such insights are vital for One Health strategies, which integrate human, animal, and environmental health approaches to mitigate the disease burden.

## 1. Introduction

Lagochilascariasis is an emerging zoonotic disease caused by the parasitic nematode *Lagochilascaris minor*. It remains a concern for health authorities worldwide, especially in tropical areas. Lagochilascariasis is underdiagnosed due to its symptoms overlapping with other diseases [1]. The order of carnivores and rodents is an important aspect of the life cycle of the parasite and the consequences of anthropization. This emphasizes the need for research covering disease-specific epidemiological features, which are pivotal for identifying the distribution and frequency in various habitats [2]. Clarifying the details of the various phases within this life cycle will provide insight into the transmissibility dynamics within impacted populations. Having a clear understanding on how these factors differ across demographics is crucial to promoting the identification and management of the disease as early as possible [3].

The other important issue is the difficulty of diagnosis experienced in detecting lagochilascariasis. Thinking about the current state of diagnostic technologies and methodologies will help us to enhance our detection and treatment of this disease. As technological capabilities continue to grow, their integration into public health systems is essential for robust management of infectious diseases. High levels of interaction between humans and wildlife, particularly in biodiversity-rich regions like the Amazon, pose a considerable risk of zoonotic disease transmission. This makes it clear just how crucial an integrated approach is to understanding the ways that environmental change affects disease cycles [4].

Lagochilascariasis is a condition in which health inequalities need to be explored at the socioeconomic level. Socioeconomic status affects health outcomes and has a significant impact on the treatment and care received, acting as a barrier. This indicates the importance of complete interventions covering these socioeconomic determinants to achieve effective disease management. Public health implications: relevant policies can guide resource allocation toward effective management and preventive strategies as well as the effectiveness of current treatment modalities, including antiparasitic medications and surgical approaches in severe cases. The One Health approach to lagochilascariasis will ensure that it is understood by people from diverse fields and will foster communication and collaboration between disciplines in order to better address the multiple facets of lagochilascariasis affecting humans, animals, and the environment [5].

Finally, this review aims to address key knowledge gaps in the understanding of lagochilascariasis, a zoonosis with substantial yet underexplored impact in the Brazilian Amazon Biome. This region presents unique ecological and socio-environmental conditions such as high biodiversity, intense anthropization, and frequent human–wildlife interactions that are likely modulating the transmission dynamics of *Lagochilascaris minor*. The southeastern region of Pará, where most reported cases occur, exemplifies how deforestation, habitat fragmentation, and socioeconomic vulnerability converge to create a high-risk context for zoonotic disease emergence [6]. By focusing on the Amazon Biome, this review not only highlights the role of wildlife in disease transmission but also illustrates how regional environmental transformations contribute to public health threats. Therefore, this narrative review provides a comprehensive synthesis that encompasses the epidemiology of lagochilascariasis, wildlife reservoirs, anthropogenic impacts, diagnostic and public health challenges, educational strategies, and future research directions ultimately contributing to a better-informed, One Health–oriented approach to managing this neglected tropical disease.

## 2. Materials and Methods

This study is a descriptive research work based on a narrative literature review according to the classification proposed by Grant and Booth [7]. This review format was chosen due to the broad and complex nature of the topic, which involves ecological, parasitological, and public health aspects of lagochilascariasis in the Brazilian Amazon Biome, particularly in the context of One Health. To ensure methodological transparency and rigor, the review incorporated systematized search elements, although it does not constitute a systematic review per se. Literature searches were conducted in multiple electronic databases, including Periódicos CAPES, PubMed, Scopus, ResearchGate, SciELO, Google Scholar, Academia.edu, BDTD, Redalyc, Science.gov, ERIC, ScienceDirect, SiBi, WorldWideScience, PePSIC, and Scholarpedia. The search strategy used combinations of the following terms: lagochilascariasis, neglected zoonosis, Brazilian Amazon Biome, epidemiology, anthropization, One Health, wildlife reservoirs, helminthiasis, and zoonotic transmission. Inclusion criteria encompassed peer-reviewed articles and academic publications that addressed the defined search terms and presented relevant information on wild animal reservoirs in the Amazon Biome and/or connections to zoonotic transmission and One Health. Exclusion criteria included publications that were not directly related to the Amazon Biome or that provided limited discussion on parasitic diseases in wildlife. Publications were evaluated based on methodological clarity, relevance to the review’s objectives, and scientific credibility. A total of 79 unique publications were identified and included. An overlap rate of 95% was observed across databases, indicating a high level of consistency in article retrieval. Due to the limited number of studies specifically addressing lagochilascariasis in wildlife and its One Health implications, all identified publications were incorporated into the final synthesis. This approach, although primarily narrative, was strengthened by its structured and reproducible search and selection methods, aiming to provide a comprehensive and integrated overview of the available scientific evidence on this emerging zoonosis in the Amazon region.

## 3. Review

### 3.1. Overview of Lagochilascariasis

Lagochilascariasis is a neglected zoonotic parasitic disease caused by the nematode *Lagochilascaris minor* (Figure 1) that affects humans and wild animals primarily in tropical and subtropical regions. This condition is particularly relevant in the Brazilian Amazon, where most cases have been reported [8]. The disease involves a complex transmission cycle, which, as detailed in Section 3.3, includes interactions between wildlife, domestic animals, and humans. Adult worms present a trilabiate anterior end and robust esophagus, with males measuring 8–14 mm and females up to 18 mm in length. Eggs are oval, measuring approximately 50–60 µm in length and 30–40 µm in width, with thick, embryonated shells. Microscopy reveals typical ascarid characteristics, such as a three-lipped anterior end and muscular esophagus [9,10]. Wild carnivores and rodents serve as the definitive and intermediate hosts, respectively [1,2,3]. Human infection occurs accidentally through the ingestion of raw or undercooked meat from infected animals [3]. However, this zoonosis remains underdiagnosed due to clinical similarities with other chronic infections and limited awareness among clinicians in endemic areas [4,5].

Deforestation and anthropogenic habitat fragmentation in the Amazon Biome have increased contact between humans and wildlife reservoirs, contributing to modifications in the disease’s epidemiology [11]. Such ecological disruptions facilitate the emergence and spread of zoonotic pathogens, particularly in regions undergoing rapid environmental change [11,12]. Environmental conditions like temperature and humidity critically influence the survival and infectivity of *L. minor* eggs [13].

The clinical presentation of lagochilascariasis is often chronic and heterogeneous, involving various anatomical regions [14,15,16]. These manifestations are influenced by host-specific factors, including age, immune response, and parasitic load [17].

Molecular diagnostic tools such as conventional and real-time PCR have significantly improved the accuracy of identifying *L. minor*. However, these resources are rarely available in remote or underserved areas where the disease is endemic [18,19]. Socioeconomic inequalities further intensify the impact of lagochilascariasis. Vulnerable populations often face limited access to health services and possess low awareness of zoonotic risks [20,21], contributing to delayed diagnosis and worse health outcomes [22,23]. The effectiveness of public health strategies is reduced when these socioeconomic and educational barriers are not addressed [24,25]. To mitigate these interconnected drivers of transmission, adopting a One Health approach is crucial, emphasizing the interconnectedness of human, animal, and environmental health [5].

### 3.2. Epidemiology of Lagochilascariasis

Understanding the epidemiology and transmission dynamics of this neglected zoonotic disease is critical for designing appropriate public health interventions [26]. Lagochilascariasis is endemic in parts of South America, especially within the Amazon region, where ecological conditions facilitate its spread (Figure 2) [27]. The disease is often underreported due to overlapping clinical pictures with other infectious diseases, making diagnosis and management challenging [28].

As detailed in Section 3.3, the epidemiology is deeply influenced by human–wildlife interactions and environmental factors like temperature and humidity, which affect the survival and development of *L. minor* eggs [29,30,31]. Changes in land use, such as deforestation and urban expansion, alter traditional habitats and influence the parasite’s distribution patterns [32,33,34]. Socioeconomic conditions play a significant role in the incidence and management of lagochilascariasis [35]. Addressing these underlying determinants is vital for effective disease control [36,37].

### 3.3. Role of Wildlife in Disease Transmission

Lagochilascariasis is a neglected parasitic disease caused by the nematode *L. minor*, whose intricate and poorly understood life cycle exemplifies a classical zoonotic interface, involving interactions among wildlife, domestic animals, the environment, and humans. Understanding each stage of this cycle is essential not only for elucidating the parasite’s ecology but also for developing targeted public health strategies and veterinary interventions within a One Health framework [31,32].

The life cycle of *L. minor* (Figure 3) [31] is heteroxenous, requiring both intermediate and definitive hosts to complete its development. It begins when carnivorous definitive hosts such as wild canids and felids (e.g., foxes, ocelots, domestic cats) consume infected intermediate hosts, primarily rodents, that harbor encysted third-stage larvae (L3) in their tissues [2,31,38]. In the gastrointestinal tract and surrounding tissues of these definitive hosts, the larvae mature into adult nematodes that localize preferentially in the nasopharynx, mastoid cells, paranasal sinuses, and cervical tissues [38,39]. Here, they reproduce and release eggs, which are eliminated in the feces into the external environment (Stage 1). Once in the environment, the eggs undergo embryonation (Stages 2–3), becoming infective under favorable conditions of temperature and humidity [31]. These infective eggs are then ingested by small mammals, especially rodents, via contaminated food or soil (Stage 4). Within these intermediate hosts, the larvae hatch, penetrate tissues, and migrate primarily to skeletal muscles and subcutaneous connective tissues, where they encyst as L3 larvae (Stage 5) [40]. Experimental studies in murine models demonstrate the development of granulomatous inflammatory reactions, often accompanied by intense eosinophilic infiltration, surrounding these larval forms [16]. Predatory carnivores close the cycle (Stages 6–8) when they feed on these infected rodents, perpetuating the parasite in the ecosystem. In humans, infection occurs incidentally, often through the consumption of raw or undercooked wild animal meat during hunting, bushmeat practices, or survival activities in forested and peri-urban regions (Stage 9) [31,32,41].

**Figure 3 tropicalmed-10-00177-f003:**
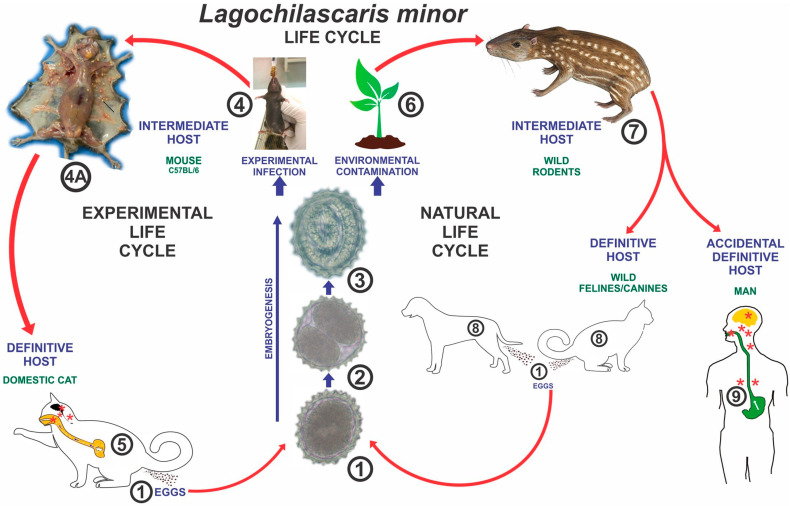
Life cycle of *Lagochilascaris minor*. Parasite eggs are eliminated from the host organism through feces (**1**), undergo division (**2**), and develop into the infecting stage (**3**). The infecting egg may be either orally inoculated into the mouse (**4**) or contaminate the environment (**6**). In experimental infection, granulomatous nodules containing third-stage larvae are observed in the muscles and subcutaneous tissue of a mouse infected with the helminth (**4A**). Experimental definitive hosts are infected through ingestion of intermediate hosts containing third-stage encysted larvae (**5**). Once in the environment (**6**), infecting eggs are ingested by wild rodents (**7**). Wild felines/canines ingest intermediate hosts containing third-stage larvae and eliminate parasite eggs into the environment through feces (**8**). Human infection originates from the ingestion of uncooked or partially cooked meat of wild rodents containing encysted larvae (**9**) [31].

The clinical presentation in definitive hosts, particularly domestic cats and dogs, is marked by chronic submandibular abscesses, oronasopharyngeal fistulas, otitis externa with polypoid formations, and in advanced cases, neurological complications due to central nervous system involvement [38,39,40,41,42]. These lesions correspond histopathologically to extensive granulomatous tissue reactions induced by both migrating larvae and adult worms [16,17,18].

Wildlife plays a central epidemiological role in maintaining the transmission cycle. Multiple species of small rodents serve as intermediate hosts, while definitive hosts include a wide range of carnivores, including foxes, domestic dogs, wild felines, and others [38,39,40,41,42]. Their ecological behaviors, such as scavenging, territorial marking, and frequent defecation near human settlements, facilitate environmental contamination with infective eggs, especially in fragmented landscapes [38,39,40].

Anthropogenic environmental changes, notably deforestation, habitat fragmentation, and unregulated land use, have intensified the contact between humans and wildlife reservoirs. As pristine forest ecosystems are degraded, wildlife is forced into closer proximity with human populations, amplifying the potential for zoonotic spillover [5,39,40]. This scenario is particularly concerning in the Amazon Biome, where biodiversity loss coincides with socioeconomic vulnerability, creating a perfect storm for the emergence and spread of zoonoses like lagochilascariasis.

In summary, the transmission dynamics of Lagochilascaris minor are shaped by a web of biological, ecological, and anthropogenic factors. Comprehensive understanding and control of lagochilascariasis demand a multidisciplinary One Health approach, integrating parasitology, wildlife ecology, public health, and socio-environmental policy [36,37,41,42].

### 3.4. Impact of Anthropization on Disease Dynamics

Anthropization, defined by urbanization, deforestation, and habitat fragmentation, significantly impacts the transmission routes of Lagochilascaris minor [43,44]. As ecosystems change, it becomes increasingly important to understand how these dynamics affect the epidemiology of lagochilascariasis [45]. Expanding urban areas and agricultural practices create suitable conditions for the parasite’s persistence [46]. Human–wildlife interactions driven by land use changes facilitate the emergence of lagochilascariasis in new geographical locations [47]. As detailed in Section 3.3, these interactions increase the possibility of outbreaks, particularly in regions with underdeveloped health infrastructure [48]. Socioeconomic factors like poverty further increase vulnerability. Integrated conservation approaches and a One Health framework, which acknowledge the interconnectedness of human, animal, and ecosystem health, are essential for developing integrative public health policies [49,50,51].

### 3.5. Public Health Implications

The complexity and multifactorial nature of the public health aspects of lagochilascariasis highlight the importance of an integrative approach that considers environmental factors, management strategies, resource distribution policies, and socioeconomic determinants of health [52,53]. Lagochilascariasis could be prevented through dedicated public health policies that could improve resource allocation to the prevention and management of the disease [54,55]. Such policies should mobilize resources toward health education campaigns aimed at the general public as well as healthcare providers to ensure that these health workers are better equipped to recognize early-stage symptoms to enable timely diagnosis of the disease [56,57]. Also, investing in effective disease monitoring systems is important to assess the incidence and spread of lagochilascariasis over time, as this information can inform public health actions and target resources [58]. This will guide the establishment of multidisciplinary health models that involve actors from different sectors to reach a collective understanding of the disease and the intervention strategies that offer the greatest potential to slow down its spread [59,60]. Effective interventions must prioritize resource allocation toward health education campaigns and improve healthcare access for vulnerable populations to reduce the disease burden [61,62]. Investing in disease monitoring systems is also critical for informed public health actions. Interdisciplinary collaboration is essential to provide pragmatic management approaches, integrating insights from veterinary research, environmental policies, and new diagnostic technologies to advance public health responses [63,64].

### 3.6. Diagnostic Challenges

Diagnosing lagochilascariasis presents a significant challenge to clinicians, particularly in endemic regions. The disease often manifests with nonspecific symptoms including fever, malaise, lymphadenopathy, and cervical masses that closely mimic fungal or bacterial infections such as paracoccidioidomycosis, histoplasmosis, or tuberculous lymphadenitis. This clinical overlap frequently results in misdiagnosis, delayed treatment, and underreporting of cases [5,37].

The histopathological features of Lagochilascaris minor infection, especially the formation of chronic granulomatous lesions, further complicate differentiation from other granulomatous diseases. Consequently, clinicians must adopt a high index of suspicion when treating patients with compatible symptoms, particularly in areas where the parasite is known to circulate [11,40,41,42].

A definitive diagnosis is traditionally achieved through the direct identification of *L. minor* eggs or adult nematodes in clinical specimens such as pus, surgical drainage, or biopsy material. Microscopy, while widely accessible, suffers from low sensitivity and may fail to detect early or mild infections [9,10].

Advancements in molecular diagnostics, including conventional PCR, real-time PCR, and sequencing of conserved ribosomal regions such as 18S and 12s rDNA, have significantly improved the specificity and sensitivity of diagnosis [37]. DNA barcoding has recently enabled identification of *L. minor* in atypical human cases [62]. These techniques are capable of detecting even low parasite loads and of distinguishing *L. minor* from other nematodes [37,62]. However, their availability remains restricted to specialized laboratories, limiting their impact in rural or underserved communities. At present, serological tests are not standardized for lagochilascariasis, and their diagnostic value remains uncertain [5,23].

Socioeconomic inequality is a key determinant of diagnostic and treatment outcomes. In resource-limited settings, individuals from low-income backgrounds often face financial constraints, geographic isolation, and limited health literacy, all of which delay healthcare-seeking behavior and restrict access to high-quality diagnostics [63,64].

These disparities create a vicious cycle: delayed diagnosis leads to disease progression, which increases the complexity and cost of treatment, further exacerbating health inequities. Therefore, public health strategies must go beyond technical solutions and incorporate equity-focused interventions that address the social determinants of health [65,66,67].

There is no universally accepted treatment protocol for lagochilascariasis. Clinical management is typically empirical and based on anecdotal evidence. Ivermectin has demonstrated effectiveness against larval forms of *L. minor* in both humans and animal models, while benzimidazole derivatives such as albendazole and levamisole are commonly used, albeit with variable clinical outcomes [10,18]. In advanced or refractory cases, surgical intervention, including abscess drainage or debridement, may be necessary. A major concern is the high recurrence rate, which underscores the importance of prolonged and combined anthelmintic therapy. However, without standardized protocols, treatment remains inconsistent and outcomes unpredictable [40,41].

Growing evidence from endemic areas points to the urgent need for a standardized diagnostic algorithm for lagochilascariasis [68,69]. Such protocols would enable clinicians to apply uniform criteria for case detection, promote early diagnosis, and facilitate comparability across surveillance systems. Moreover, integrating diagnostic findings with epidemiological surveillance, especially in the context of One Health frameworks, could provide valuable insights into the spatial dynamics, transmission cycles, and ecological drivers of the disease [70,71,72,73]. Incorporating both genomic (e.g., parasite sequencing) and non-genomic data (e.g., host ecology, land use) offers a promising avenue for building evidence-based public health responses [5].

### 3.7. Educational Strategies for Public Awareness

Educational strategies for lagochilascariasis must be culturally relevant, community-focused, and designed to effectively disseminate knowledge in endemic areas [70]. Public health education should be engaging, collectively designed, and tailored to the local context. Community health workers play a pivotal role in spreading information about prevention and treatment directly within communities [71]. Their familiarity with local beliefs and practices enables them to develop health messages that resonate with the community [72]. Creating learning materials that incorporate local cultural knowledge is essential. This can include content in local dialects and culturally relevant explanations of disease transmission. Educational campaigns should leverage local cultural strengths to promote effective preventative and health behaviors [70,71,72].

Technological and social media tools are powerful resources for public awareness. They provide the opportunity to reach diverse audiences quickly and facilitate discussions across cultural boundaries [73]. Social media platforms, in particular, can engage younger age groups and encourage healthy behaviors through personalized messages such as informational videos and infographics [74,75]. Peer education models are another promising approach. Training community members as peer educators fosters a sense of ownership over community health and helps spread information on testing and resources, reducing the stigma associated with the disease [5,41]. To sustain educational programs, public health policies must prioritize long-term investments in education, outreach, and community engagement [43]. Partnerships between public health and educational organizations supported by policymakers can enhance the community capacity to address health risks [46]. Involvement of local leaders in designing and delivering health education can further amplify its impact, as these leaders shape community beliefs and practices [53].

### 3.8. Research Gaps and Future Directions

There are significant research gaps in understanding lagochilascariasis [1], particularly in its epidemiology, clinical implications, and socioeconomic contexts. Targeted research initiatives are needed to optimize prevention interventions and inform public health measures in endemic settings [2]. Despite advances, major gaps persist regarding the complete ontogeny and immunopathogenesis of *L. minor* in natural hosts. Most knowledge is extrapolated from experimental murine and feline models, which may not fully reflect wild host dynamics. Future research should prioritize the standardization of diagnostic protocols; mapping of wild host species and their ecological roles; immunological studies on host–parasite interaction; and the impact of anthropogenic change on parasite distribution. Collaboration between veterinary parasitologists, ecologists, and public health professionals is imperative to expand the One Health understanding of this emerging zoonosis [76,77,78,79,80]. Key areas for future research include technological advancements in diagnostics, the socioeconomic impacts on disease management, and interdisciplinary collaboration [5]. Climate change data must be integrated into research to understand how fluctuations affect the distribution of *L. minor* and its hosts [31]. Understanding the impact of temperature and precipitation changes on parasite life cycles and host interactions is crucial for developing proactive public health interventions [27]. Data-driven interventions should also address chronic socioeconomic disparities that drive health inequities [21]. The One Health framework should be refined to address emerging zoonoses like lagochilascariasis in dynamic ecosystems [42]. Research initiatives must promote partnerships among public health officials, veterinarians, environmental scientists, and policymakers to coordinate responses to zoonotic disease emergence. Integrative approaches will enhance preparedness and adaptability against future health threats [72,75]. Understanding the biology of *L. minor* is not only fundamental for diagnosis and treatment but also for identifying environmental conditions (e.g., humidity, host abundance) that sustain its transmission, especially in Amazonian regions undergoing deforestation and anthropization [5].

## 4. Conclusions

Lagochilascariasis exemplifies how environmental change, wildlife ecology, and socioeconomic factors converge to shape zoonotic disease dynamics in the Amazon Biome. The evidence reviewed suggests that anthropogenic pressures such as deforestation, habitat fragmentation, and increased human–wildlife contact are not only altering the transmission patterns of *L. minor* but also amplifying the vulnerability of rural and marginalized populations. Moving forward, control strategies must go beyond conventional medical approaches and incorporate land-use planning, wildlife surveillance, and socioeconomic development. The application of a One Health framework is essential to integrate these elements and support coordinated actions across sectors. Future research should prioritize mapping wildlife reservoirs, improving early diagnosis through accessible molecular tools, and evaluating the impact of climate variability on transmission risk. Strengthening community engagement and tailoring interventions to local sociocultural contexts will be key to enhancing disease awareness and prevention efforts. Lagochilascariasis, although neglected, offers a unique lens to understand the broader patterns of zoonotic emergence under environmental disruption. Addressing it effectively requires interdisciplinary collaboration and long-term commitment to ecological and social determinants of health.

## Figures and Tables

**Figure 1 tropicalmed-10-00177-f001:**
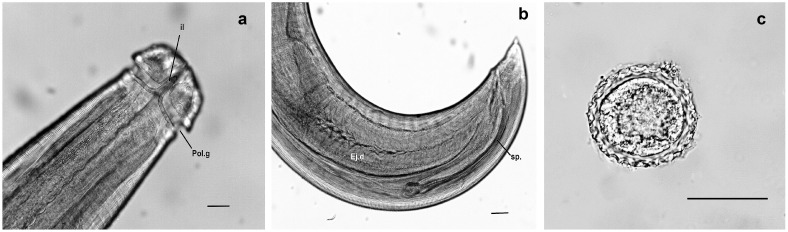
(**a**) Anterior end of *Lagochilascaris minor* (lateral view): Post-labial groove (Pol.g) and interlabia (il). Bar = 0.04 mm. (**b**) Posterior end (lateral view) of male showing the ejaculatory duct (Ej.d) and spicule (sp.). Bar = 0.05 mm. (**c**) Egg. Bar = 0.05 mm [8].

**Figure 2 tropicalmed-10-00177-f002:**
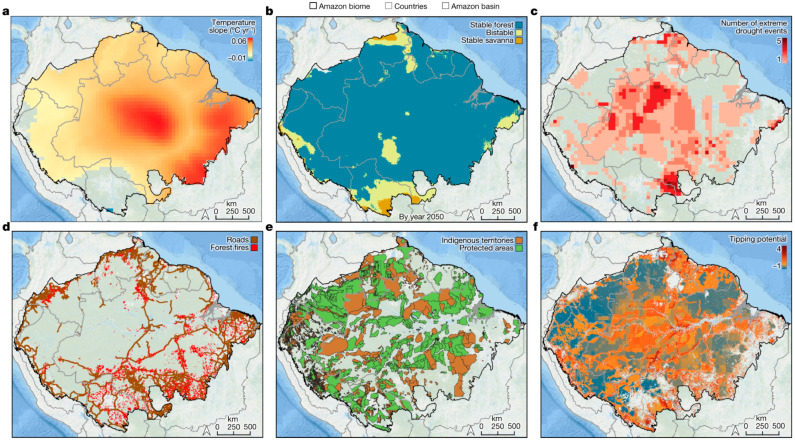
(**a**) Changes in the dry season (July–October) mean temperature reveal widespread warming, estimated using simple regressions between time and temperature observed between 1981 and 2020 (with *p*  <  0.1). (**b**) Potential ecosystem stability classes estimated for year 2050, adapted from current stability classes (Extended Data Figure 1b) by considering only areas with significant regression slopes between time and annual rainfall observed from 1981 through 2020 (with *p*  <  0.1) (see Extended Data Figure 3 for areas with significant changes). (**c**) Repeated extreme drought events between 2001 and 2018 (**d**) Road network from where illegal deforestation and degradation may spread. (**e**) Protected areas and Indigenous territories reduce deforestation and fire disturbances. (**f**) Ecosystem transition potential (the possibility of forest shifting into an alternative structural or compositional state) across the Amazon Biome by the year 2050 inferred from compounding disturbances (**a**–**d**) and high-governance areas (**e**) excluding accumulated deforestation until 2020 and savannas. Transition potential rises with compounding disturbances and varies as follows: less than 0 (in blue) as low; between 1 and 2 as moderate (in yellow); more than 2 as high (orange–red). Transition potential represents the sum of (1) slopes of dry season mean temperature (as in (**a**), multiplied by 10); (2) ecosystem stability classes estimated for year 2050 (as in (**b**)), with 0 for stable forest, 1 for bistable, and 2 for stable savanna; (3) accumulated impacts from extreme drought events, with 0.2 for each event; (4) road proximity as a proxy for degrading activities, with 1 for pixels within 10 km from a road; (5) areas with higher governance within protected areas and Indigenous territories, with −1 for pixels inside these areas [27].

## Data Availability

No new data were created or analyzed in this study. Data sharing is not applicable to this article.

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
