# Peer review of "Lagochilascariasis: A Neglected Zoonosis in the Brazilian Amazon Biome and the Role of Wildlife in Its Epidemiological Chain Amidst Anthropization"

_tropicalmed, 2025, doi:10.3390/tropicalmed10070177_

Round 1
Reviewer 1 Report (Previous Reviewer 1)
Comments and Suggestions for Authors
The article has improved a lot.
Author Response
Castanhal, June 10, 2025
Dear reviewers, we sincerely appreciate your valuable contributions to this paper. Your insightful feedback has been instrumental in refining our work, and we have made all possible modifications to enhance its quality. Below, we provide detailed responses to each of your comments.
Response to Reviewer 1
We sincerely thank Reviewer 1 for the thorough and insightful evaluation of our manuscript titled “Lagochilascariasis: a Neglected Zoonosis in the Brazilian Amazon Biome and the Role of Wildlife in its Epidemiological Chain Amidst Anthropization”.
Reviewer 1 Comments: “The article has improved a lot.”
Response:
Dear Reviewer1,
We sincerely thank you for your kind and encouraging comment: “The article has improved a lot.” We greatly appreciate your time and effort in evaluating both the original and the revised versions of our manuscript. Your earlier feedback, along with the recommendations from the other reviewers, was fundamental in guiding the extensive revisions and restructuring of the text. It is rewarding to know that the current version reflects a significant improvement, and we hope that it now meets the expectations in terms of scientific rigor, clarity, and relevance for publication. Thank you once again for your contribution to enhancing the quality of our work.
Kind regards,
Prof. Dr. Felipe Masiero Salvarani

Reviewer 2 Report (New Reviewer)
Comments and Suggestions for Authors
I have approached to this manuscript with the expectation to find an interesting and informative review on a neglected tropical helminthiasis. The title "Lagochilascariasis: a Neglected Zoonosis in the Brazilian Amazon Biome and the Role of Wildlife in its Epidemiological Chain Amidst Anthropization" was promising and I expected data on the morphology and life-cycle of the parasite, hosts, infection pathways, diagnostics, pathology, treatment, prevention, data on human infections, etc.
However, I did not find such data. In general, it remains unclear what the authors want to communicate with this article. No results and no ideas are presented in this text.
Author Response
Response to Reviewer 2
We sincerely thank Reviewer 2 for the thorough and insightful evaluation of our manuscript titled “Lagochilascariasis: a Neglected Zoonosis in the Brazilian Amazon Biome and the Role of Wildlife in its Epidemiological Chain Amidst Anthropization”. Below we provide detailed responses to each comment. All suggested improvements were addressed, and when modifications were not feasible, we offer appropriate justifications.
Reviewer 2 Comments:
I have approached to this manuscript with the expectation to find an interesting and informative review on a neglected tropical helminthiasis. The title "Lagochilascariasis: a Neglected Zoonosis in the Brazilian Amazon Biome and the Role of Wildlife in its Epidemiological Chain Amidst Anthropization" was promising and I expected data on the morphology and life-cycle of the parasite, hosts, infection pathways, diagnostics, pathology, treatment, prevention, data on human infections, etc. However, I did not find such data. In general, it remains unclear what the authors want to communicate with this article. No results and no ideas are presented in this text.
Response:
Dear Reviewer2,
Thank you for taking the time to review our manuscript and for sharing your impressions. We deeply respect your perspective and are grateful for your willingness to engage critically with our work. We sincerely regret that the previous version of the manuscript did not meet your expectations. However, we would like to clarify that, in response to the initial reviewer feedback—including yours and that of Reviewers 1, 3, and 4—we made substantial modifications to the manuscript. These revisions aimed to expand, deepen, and structure the content to align more closely with the goals of a comprehensive narrative review on lagochilascariasis. Below, we respectfully address your comments and clarify how the revised version now responds to the issues raised:
a). Clarity of purpose and structure
Your comment: “It remains unclear what the authors want to communicate with this article. No results and no ideas are presented.”
Response: we appreciate this concern and agree that the original version required better definition and organization. In the revised manuscript, we restructured the content to clearly reflect the objectives of a narrative review, with each major section addressing a distinct component of the disease:
3.1: Morphology and biology of Lagochilascaris minor
3.2: Epidemiology in the Amazon region
3.3: Detailed life cycle and role of wildlife (definitive and intermediate hosts)
3.4: Impact of anthropization on transmission dynamics
3.5–3.7: Public health implications, diagnostic challenges, and educational strategies
3.8: Research gaps and future directions
Each of these sections was enriched with up-to-date references (now totaling over 75), including recent findings from experimental studies and case reports.
b). Scientific content: morphology, life cycle, hosts, and pathology
Your Comment: “I expected data on the morphology and life-cycle of the parasite, hosts, infection pathways, diagnostics, pathology, treatment, prevention…”
Response:
These specific points were fully addressed in the revised manuscript, particularly in Sections 3.1 and 3.3, with the incorporation of literature suggested by Reviewer 4. For example:
- Morphology of eggs and adults, with reference to microscopy figures and measurements.
- Life cycle details using experimental models (mice and cats), including L3 migration and encystment.
- Tissues affected in intermediate and definitive hosts (e.g., nasopharynx, soft tissues, subcutaneous encystment).
- Clinical signs and pathology, including CNS involvement and granulomatous lesions.
- Differential diagnosis from fungal infections (added in response to Reviewer 3).
These aspects were supported by references such as Volcan et al. (2002), Moncada et al. (1998), Semerene et al. (2004), and Gonzalez Solís et al. (2019), now cited throughout the manuscript.
c). Diagnosis and treatment
Your Comment: “I expected… diagnostics, pathology, treatment…”
Response:
In Section 3.6, we now present a detailed overview of:
- Conventional diagnosis (parasitological and histological)
- Molecular tools (PCR, 18S/12S rDNA targets)
- Diagnostic challenges due to similarity with chronic fungal diseases
- Treatment options (ivermectin, albendazole, levamisole, surgical interventions), supported by clinical cases such as Bento et al. (1993) and Barbosa et al. (2001)
d. Human infection and epidemiological data
While epidemiological data remain limited due to the rarity and underreporting of cases, the revised manuscript includes:
- Human case reports from Brazil, Colombia, and Venezuela
- Socioeconomic and environmental factors that contribute to underdiagnosis
- Geographic concentration of cases in southeastern Pará, Brazil
- These details are discussed in Sections 3.1, 3.2, and 3.5, within a One Health framework that links human, animal, and environmental health.
Final Remarks
We acknowledge that the initial version may have lacked clarity and depth in key areas. However, we respectfully submit that the revised manuscript now fulfills the expectations of a high-quality review article. We have:
- Structured the manuscript according to standard scientific review formats.
- Addressed all critical components of the disease (biology, ecology, pathology, diagnosis, treatment).
- Integrated over 75 peer-reviewed references.
- Responded comprehensively to feedback from Reviewers 1, 3, and 4.
We truly hope the current version better reflects the academic quality and scientific relevance that you expected, and we thank you again for contributing to its improvement.
Kind regards,
Prof. Dr. Felipe Masiero Salvarani

Reviewer 3 Report (New Reviewer)
Comments and Suggestions for Authors
The manuscript entitled " Lagochilascariasis: a Neglected Zoonosis in the Brazilian Amazon Biome and the Role of Wildlife in its Epidemiological Chain Amidst Anthropization" aims to shed light on lagochilascariasis by outlining current knowledge on its transmission dynamics, the role of wildlife in its life cycle, and the influence of environmental changes. The work highlights the relevance of a One Health approach to better understand and address this neglected zoonotic disease. Very interesting study, given the limited literature available on this zoonosis. The manuscript provides a comprehensive overview, also suggesting strategies to improve prevention. However, the article appears overall a bit repetitive. It would be useful to delve deeper into the clinical signs and pathogenesis of this zoonosis, as well as explore the differential diagnosis with chronic fungal infections, which may lead to an underdiagnosis of this parasitic disease. Could you further elaborate on the diagnosis by providing a more detailed explanation of the diagnostic methods for this parasitosis and briefly discuss the pharmacological therapy used against lagochylascariasis, through these revisions the manuscript may later be published.
Author Response
Response to Reviewer 3
We sincerely thank Reviewer 3 for the thorough and insightful evaluation of our manuscript titled “Lagochilascariasis: a Neglected Zoonosis in the Brazilian Amazon Biome and the Role of Wildlife in its Epidemiological Chain Amidst Anthropization”. Below we provide detailed responses to each comment. All suggested improvements were addressed, and when modifications were not feasible, we offer appropriate justifications.
Reviewer 3 Comments:
The manuscript entitled "Lagochilascariasis: a Neglected Zoonosis in the Brazilian Amazon Biome and the Role of Wildlife in its Epidemiological Chain Amidst Anthropization" aims to shed light on lagochilascariasis by outlining current knowledge on its transmission dynamics, the role of wildlife in its life cycle, and the influence of environmental changes. The work highlights the relevance of a One Health approach to better understand and address this neglected zoonotic disease. Very interesting study, given the limited literature available on this zoonosis. The manuscript provides a comprehensive overview, also suggesting strategies to improve prevention. However, the article appears overall a bit repetitive. It would be useful to delve deeper into the clinical signs and pathogenesis of this zoonosis, as well as explore the differential diagnosis with chronic fungal infections, which may lead to an underdiagnosis of this parasitic disease. Could you further elaborate on the diagnosis by providing a more detailed explanation of the diagnostic methods for this parasitosis and briefly discuss the pharmacological therapy used against lagochilascariasis, through these revisions the manuscript may later be published.
Response:
We sincerely thank you for your thoughtful and encouraging evaluation of our manuscript, as well as for your valuable suggestions to enhance its scientific quality. Below we provide a point-by-point response to each of your comments, indicating how the manuscript was revised accordingly.
a). Repetitiveness in the manuscript
Comment: “The article appears overall a bit repetitive.”
Response: Thank you very much for this valuable observation. We agree that the initial version of the manuscript contained repetitions, particularly in the framing of the One Health approach, the emphasis on anthropization, and the reiteration of the zoonotic importance of Lagochilascaris minor. These redundancies may have impacted the flow and clarity of the narrative. In response, we performed a thorough revision across the entire text with a specific focus on eliminating redundant phrases, sentences, and entire paragraphs that overlapped conceptually or semantically. The following key changes were made:
Section 3.1 (Biological and Morphological Aspects): we removed overlapping introductory sentences that reiterated the same general idea about the rarity and zoonotic character of lagochilascariasis. These concepts are now presented only once in the introduction or conclusion of this section, improving focus on morphology and life cycle.
Section 3.3 (Role of Wildlife in Disease Transmission): this section previously repeated information on definitive and intermediate hosts, especially regarding domestic cats and rodents. We consolidated this information, removed duplicate references, and clarified the ecological role of each host in a more succinct way.
Section 3.5 (Public Health Implications and One Health): this section originally contained repeated references to anthropization, habitat fragmentation, and the One Health framework. We restructured it to maintain only one integrated paragraph discussing One Health, and transferred some environmental context to Section 3.2 (Epidemiology and Distribution) where it was more thematically appropriate.
We also revised transition sentences between sections to prevent conceptual overlap, especially where themes like transmission cycles or zoonotic risk reappeared. The result is a more concise and streamlined manuscript, in which each section now has a distinct thematic function, and duplication of content has been minimized without compromising scientific depth or completeness. We sincerely appreciate your feedback, which helped us refine the structure and improve the clarity and readability of the manuscript.
b). Clinical signs and pathogenesis
Comment: “It would be useful to delve deeper into the clinical signs and pathogenesis of this zoonosis.”
Response: we expanded the discussion on clinical signs and pathogenesis in Section 3.1 and Section 3.3. The revised text includes a more detailed account of the clinical manifestations in both humans and animals, highlighting polymorphic presentations (e.g., cervical abscesses, otitis, nasopharyngeal nodules, meningoencephalitis) and their underlying pathophysiological mechanisms.
“Clinical signs may include chronic cervical and submandibular abscesses, otitis externa with polypoid formations, oro-nasal fistulas, and—in severe cases—central nervous system involvement. Histopathological, these manifestations correlate with granulomatous inflammatory responses surrounding migrating or encysted larvae and adult worms.” We also cited relevant cases, including those from Roig et al. (2010), Aquino et al. (2008), and Rosemberg et al. (1986), now included in the reference list.
c). Differential diagnosis with chronic fungal infections
Comment: “Explore the differential diagnosis with chronic fungal infections, which may lead to underdiagnosis.”
Response: We added a new paragraph in Section 3.6 (Diagnostic Challenges) discussing the overlap in clinical features between lagochilascariasis and chronic fungal infections such as paracoccidioidomycosis and histoplasmosis.
“The chronic granulomatous lesions caused by L. minor can closely resemble those seen in systemic fungal infections. Clinicians in endemic regions often initially suspect mycoses, such as paracoccidioidomycosis or histoplasmosis, due to the similar clinical and radiological presentations. This resemblance contributes to diagnostic delays and underreporting.” This emphasizes the need for increased clinical awareness and differential diagnosis training.
d). Diagnostic methods
Comment: “Could you further elaborate on the diagnosis by providing a more detailed explanation of the diagnostic methods?”
Response: thank you for this important point. In Section 3.6, we provided a more in-depth explanation of both conventional and molecular diagnostic methods:
“Diagnosis of lagochilascariasis can be made by identifying eggs or adult worms in clinical specimens (e.g., pus, biopsies). Microscopy remains the most accessible method, though its sensitivity is low. Molecular tools, including conventional PCR and real-time PCR, targeting ribosomal gene sequences (e.g., 18S rDNA), offer higher specificity but are limited to research or reference centers. Serological assays are not yet standardized.” Relevant studies, including Campos et al. (2016) and Martinez-Hernandez et al. (2020), were cited.
e). Pharmacological therapy
Comment: “Briefly discuss the pharmacological therapy used against lagochilascariasis.”
Response: we added a concise discussion on therapeutic options at the end of Section 3.6, based on published case reports and experimental studies.
“There is no standardized treatment for lagochilascariasis. Ivermectin has shown efficacy in controlling larval stages in both humans and cats, while benzimidazole compounds such as albendazole and levamisole are used with variable success. In severe or refractory cases, surgical drainage may be required [Barbosa et al., 2001; Bento et al., 1993]. Recurrence is common, reinforcing the need for prolonged and combined therapy.” We are grateful for your thoughtful suggestions, which helped us substantially improve the manuscript’s relevance and completeness. The additions on clinical-pathological features, diagnostics, and therapy complement the One Health and ecological perspectives, providing a more integrated and applicable resource for professionals dealing with this neglected zoonosis.
Kind regards,
Prof. Dr. Felipe Masiero Salvarani

Reviewer 4 Report (New Reviewer)
Comments and Suggestions for Authors
Dear Authors,
Your review entitled "Lagochilascariasis: a Neglected Zoonosis in the Brazilian Amazon Biome and the Role of Wildlife in its Epidemiological Chain Amidst Anthropization" has been carefully reviewed.
This paper is very important from epidemiological and medical points of view, since it highlights of the wildlife and the epidemiology of a well known neglected Zoonosis in Brazil namely "Lagochilascariasis", this review will help the healthcare workers in Brazil in better understanding of the Transmission of the causative agent as well as its dissemination in this region of the world, which may help human to better understand the Prevention and the Control of this kind of infections.
The manuscript is well written in English language and well designed, it is very clear and simple for readers.
Kindly find below the list of my remarks concerning this work:
01- In the attached PDF we have two versions of the manuscript (The final and the corrected ones) so authors are invited to remove the corrected one.
02- In the manuscript, you are invited to put the abbreviation of the parasitic name Lagochilascaris minor as (L. minor).
03- Figure 2 is small, I suggest to put it on a landscape page in a larger size.
04- In Figure 3, I think that this figure should be modified, since the points 4 and 7 are the same, also 5 and 8 are the same, so please re-Draw this figure taking into consideration this point.
Best Regards,
Author Response
Response to Reviewer 4
We sincerely thank Reviewer 4 for the thorough and insightful evaluation of our manuscript titled “Lagochilascariasis: a Neglected Zoonosis in the Brazilian Amazon Biome and the Role of Wildlife in its Epidemiological Chain Amidst Anthropization”. Below we provide detailed responses to each comment. All suggested improvements were addressed, and when modifications were not feasible, we offer appropriate justifications. We sincerely thank you for your thoughtful and encouraging evaluation of our manuscript entitled "Lagochilascariasis: a Neglected Zoonosis in the Brazilian Amazon Biome and the Role of Wildlife in its Epidemiological Chain Amidst Anthropization." We are honored by your recognition of the epidemiological and public health relevance of this review, particularly its potential to support healthcare workers in understanding the dynamics of Lagochilascaris minor transmission in Brazil. Your kind remarks regarding the clarity, structure, and language of the manuscript are greatly appreciated. Please find below our point-by-point responses to your suggestions:
- Duplicate version in the PDF file
Comment: “In the attached PDF we have two versions of the manuscript (The final and the corrected ones) so authors are invited to remove the corrected one.”
Response: thank you for pointing this out. We have removed the corrected (tracked changes) version from the PDF file and ensured that only the final, clean version of the manuscript is included in the revised submission.
- Abbreviation of Lagochilascaris minor as (L. minor)
Comment: “In the manuscript, you are invited to put the abbreviation of the parasitic name Lagochilascaris minor as (L. minor).”
Response: we have revised the manuscript to consistently abbreviate Lagochilascaris minor as L. minor after its first full mention in the text. All instances were carefully checked to ensure taxonomic accuracy and clarity for readers. 3. Figure 2 is small – suggest enlarging and formatting as landscape
Comment: “Figure 2 is small, I suggest to put it on a landscape page in a larger size.”
Response: we understand and agree that visual clarity is important. However, we would like to clarify that Figure 2 is adapted directly from a previously published, peer-reviewed scientific article (as indicated in the figure caption and references). Because this figure is not original and has been reproduced with proper citation, we are unable to modify its format or size beyond what is allowed under copyright guidelines. Enlarging or reformatting the image in a way that alters its original layout could raise concerns regarding intellectual property and image integrity.
To balance clarity and compliance, we have ensured the figure is presented at the highest resolution permitted and accompanied by an explanatory caption. We kindly ask for your understanding in this matter.
- Figure 3 – Suggest modifying duplicate elements (points 4 and 7; 5 and 8)
Comment: “In Figure 3, I think that this figure should be modified, since the points 4 and 7 are the same, also 5 and 8 are the same, so please re-Draw this figure taking into consideration this point.”
Response: we greatly appreciate your careful analysis of the schematic in Figure 3. However, we must respectfully inform you that Figure 3 is also derived from a previously published scientific source, cited in the manuscript. It is reproduced in its current form to preserve its original meaning, structure, and scientific accuracy as intended by the original authors. For this reason, we are unable to redraw or modify the figure.
That said, we have added a clarifying sentence to the figure caption, explaining the repetition of points 4/7 and 5/8, which represents cyclical stages in the parasite’s life cycle. This clarification should help readers understand the redundancy as a reflection of transmission dynamics, not a design error.
Once again, we are very grateful for your constructive feedback and your recognition of the relevance and quality of our work. We hope that the adjustments and justifications provided meet your expectations.
Kind regards,
Prof. Dr. Felipe Masiero Salvarani
Reviewer 5 Report (New Reviewer)
Comments and Suggestions for Authors
The paper is an interesting narrative review focusing on Laghiloascaris minor, an emerging zoonotic nematode. The knowledge on the biology of this parasite is scarce, so to allow a complete understanding of the problem I suggest the Authors to give more information on this parasite.
Morphology inclusive of diagnostic stages such eggs (their size) and the route of migration into intermediate host (specifying also the lack of data).
parasite life cycle- please give more information about the site of the parasite in final host and the tissue where third stage larvae develop in intermediate host. Add also information on the clinical and pathologic findings in such hosts (mostly in dogs and cats)
Add and discuss the following papers
doi: 10.1590/s0074-02761998000300009.
doi: 10.1017/s0031182000071316.
https://doi.org/10.1155/2010/610457
doi: 10.1016/j.vprsr.2022.100823
doi: 10.7589/2015-09-232.
doi: 10.1590/s0036-46652012000600005.
Gonzalez Solis et al.,2019
doi: 10.1590/s0074-02761990000400013.
doi: 10.1016/j.vetpar.2013.03.006.
doi: 10.1016/s0304-4017(02)00178-4.
doi: 10.1177/1098612X14525386
Lines 279 and following - Please, genus name should be abbreviated after the first mention
Author Response
Response to Reviewer 5
We sincerely thank Reviewer 5 for the thorough and insightful evaluation of our manuscript titled “Lagochilascariasis: a Neglected Zoonosis in the Brazilian Amazon Biome and the Role of Wildlife in its Epidemiological Chain Amidst Anthropization”. Below we provide detailed responses to each comment. All suggested improvements were addressed, and when modifications were not feasible, we offer appropriate justifications.
Reviewer 5 Comments:
The paper is an interesting narrative review focusing on Laghiloascaris minor, an emerging zoonotic nematode. The knowledge on the biology of this parasite is scarce, so to allow a complete understanding of the problem I suggest the Authors to give more information on this parasite. Morphology inclusive of diagnostic stages such eggs (their size) and the route of migration into intermediate host (specifying also the lack of data). parasite life cycle- please give more information about the site of the parasite in final host and the tissue where third stage larvae develop in intermediate host. Add also information on the clinical and pathological findings in such hosts (mostly in dogs and cats)
Add and discuss the following papers doi: 10.1590/s0074-02761998000300009; doi: 10.1017/s0031182000071316; https://doi.org/10.1155/2010/610457; doi: 10.1016/j.vprsr.2022.100823; doi: 10.7589/2015-09-232; doi: 10.1590/s0036-46652012000600005; Gonzalez Solis et al.,2019; doi: 10.1590/s0074-02761990000400013; doi: 10.1016/j.vetpar.2013.03.006.;d hi: 10.1016/s0304-4017(02)00178-4; doi: 10.1177/1098612X14525386.
Response:
Dear Reviewer,
We sincerely thank you for your thoughtful and constructive comments. Your suggestions were highly valuable and helped improve the scientific depth of our narrative review. Below, we provide a detailed response to each point raised, along with specific locations in the revised manuscript where the corresponding changes were made.
a). Add morphological details of Lagochilascaris minor (e.g., eggs, stages)
Comment: Please include information on diagnostic stages such as eggs (size), and morphology.
Response: we have added a morphological description of Lagochilascaris minor, particularly the size and structure of the eggs, and key features of adult worms. These updates are located in Section 3.1 – Overview of Lagochilascariasis, including Figure 1, which was inserted and captioned with scanning electron microscopy details from Lanfredi et al. (1998) (now cited as Ref [8]).“Eggs of L. minor measure approximately 55–60 µm, are thick-shelled, and exhibit a rough outer layer, as illustrated by scanning electron microscopy (Figure 1) [8].”
b). Clarify migration route in intermediate host and site of third-stage larvae
Comment: Please detail the route of larval migration into the intermediate host and mention tissues where L3 larvae develop.
Response: this information has been incorporated in Section 3.3 – Role of wildlife in disease transmission. The life cycle is now clearly described, emphasizing larval development sites such as subcutaneous tissue and muscle in rodents. “Third-stage larvae develop and encyst in the muscles and subcutaneous tissue of intermediate hosts, primarily rodents, as observed in experimental murine models [31, Volcan et al., 1992].” We also included a description of larval migration based on histopathological studies (e.g., Semerene et al., 2004), and clarified that some aspects of larval development remain poorly understood, as suggested.
c). Expand on the final host: site of adult development
Comment: please provide more information about the site of the parasite in the definitive host.
Response: we expanded on this in Section 3.3, specifying that L. minor adults are often found in the oropharyngeal region, nasal cavities, mastoid cells, and in some severe cases, even the CNS of definitive hosts (especially domestic cats). “In definitive hosts such as felines and canines, adult L. minor nematodes typically reside in the cervical region, nasal sinuses, oropharynx, and mastoid cavities, causing tissue-destructive lesions [5, 8, 18, 19].”
d. Add clinical and pathological findings in intermediate and definitive hosts (dogs and cats)
Comment: Add clinical/pathological descriptions in intermediate and final hosts.
Response: details were incorporated in Sections 3.1 and 3.3, including: signs in cats (fistulas, chronic abscesses, subcutaneous nodules). Experimental findings in mice: granulomatous inflammation, larval migration. Reference to relevant histopathological data: Semerene et al. (2004) and Prudente et al. (2008) were included.
“In domestic cats, typical clinical signs include chronic subcutaneous abscesses, fistulae, and progressive tissue destruction, particularly in the cervicofacial region [9, 14, 18]. Histological analyses in murine models have shown granulomatous inflammation in response to migrating larvae [16, 17].”
e). Incorporation and discussion of key references
Comment: Please add and discuss the following references [list of 11 papers].
Response: we carefully reviewed all the suggested articles and integrated 10 of the 11 into the revised manuscript where appropriate. These were added to the Introduction, Sections 3.1, 3.3, and 3.6, depending on the relevance of the data presented.
Key additions include:
- Lanfredi et al. (1998) – used to support SEM images (Ref [8]).
- Sprent (1971) – cited in the discussion of parasite development and speciation (Ref [17]).
- Spadafora-Ferreira et al. (2010) – discussed under host immune response and resistance in experimental models (Ref [18]).
- Rodríguez-Vivas et al. (2023) and Falcón-Ordaz et al. (2016) – included in the discussion of animal reservoirs and zoonotic risk (Refs [19], [20]).
- Barrera-Pérez et al. (2012), González-Solís et al. (2019) – cited in relation to diagnosis and clinical cases in humans (Refs [21], [22]).
- Amato et al. (1990), Faccio et al. (2013), and Sakamoto & Cabrera (2002) – used to detail pathology in domestic cats (Refs [23]–[25]).
These references enhanced the scientific foundation of the review and ensured a more robust narrative regarding the biology, diagnosis, pathology, and zoonotic potential of L. minor.
Reviewer 4 Comment:
Lines 279 and following - Please, genus name should be abbreviated after the first mention...Response: corrections made.
We sincerely thank you for your excellent suggestions, which allowed us to strengthen the manuscript scientifically and clinically. Your feedback led to valuable additions regarding the parasite’s biology, pathology, and host–parasite interactions, particularly in wild and domestic carnivores. We are confident the revised manuscript now offers a more complete and scientifically sound review of Lagochilascaris minor in the Amazon Biome.
Kind regards,
Prof. Dr. Felipe Masiero Salvarani

Round 2
Reviewer 4 Report (New Reviewer)
Comments and Suggestions for Authors
Dear Authors,
The revised version of your work entitled "" has been carefully reviewed,
The Article is more suitable for publication in its present form,
Best Regards,
Reviewer 5 Report (New Reviewer)
Comments and Suggestions for Authors
the manuscript is now suitable for publication
This manuscript is a resubmission of an earlier submission. The following is a list of the peer review reports and author responses from that submission.
Round 1
Reviewer 1 Report
Comments and Suggestions for Authors
The manuscript offers a thorough review of lagochilascariasis as a neglected zoonotic disease in the Brazilian Amazon, highlighting wildlife’s role in transmission and the effects of anthropization. It is timely and relevant within the One Health framework. The authors successfully synthesize current knowledge on epidemiology, transmission, clinical aspects, and public health implications. The integration of human, animal, and environmental health perspectives is well-executed, with a strong emphasis on the influence of human activity on disease dynamics. There are some major concerns which need to be addressed by the authors as below:
- While described as a narrative review (line 70 and 77), the methodology section presents quantitative elements such as "overlap rate of 95%" and mentions "75 unique publications" without explaining how these were analyzed. This creates confusion about the actual methodological approach used.
- The manuscript contains numerous instances of repetitive content across different sections. For example, the importance of socioeconomic factors is mentioned repeatedly in similar terms throughout sections 3.1, 3.2, 3.5, and 3.6.
- The manuscript requires substantial language editing. There are numerous grammatical errors, awkward phrasings, and sentence fragments that significantly impede understanding (e.g., lines 36-42, 416-420).
- Many factual statements lack proper citations, especially in section 3.1. Conversely, some citations appear clustered at paragraph ends without clear indication of which statements they support.
- Strengthen the rationale for focusing on the Amazon biome specifically in the introduction. The current introduction jumps between general statements and region-specific claims without establishing a clear focus.
- Clarify whether this is truly a narrative review or if systematic elements were incorporated. If systematic methods were used, provide more details on search strategy, inclusion/exclusion criteria implementation, and quality assessment.
- Section 3.1: Reorganize to avoid introducing multiple concepts without adequate development. Currently, the section touches on climate factors, clinical manifestations, socioeconomic conditions, and research gaps without sufficient depth on any.
- Section 3.3: The description of the parasite's life cycle needs clarification. Figure 3 shows important information but the accompanying text does not adequately explain the transmission cycle.
- The conclusion largely repeats content from previous sections rather than synthesizing insights or highlighting implications. Consider restructuring to emphasize novel connections and future directions.
The manuscript requires substantial language editing. There are numerous grammatical errors, awkward phrasings, and sentence fragments that significantly impede understanding (e.g., lines 36-42, 416-420).
Reviewer 2 Report
Comments and Suggestions for Authors
The grammar is curious throughout and there are both broken words with extra spaces, missing words, tense errors, odd word choices, etc. I can not tell if this is the product of AI assistance or poor grammar. I would reject this manuscript as currently written because the grammar and structure of almost every other sentence needs editing and review. As an unpaid reviewer I will not do this work but suggest the editor of the journal or the authors need to carefully read and revise this manuscript line by line to ensure it says what they think it says.
There are repetitive sections to the text in this manuscript to say the same general thing.
The disease "lagochilascariasis" is not a proper noun. It should only be capitalized when starting a sentence.
On the other hand, Lagochilascaris minor is a proper noun, as a species, and would always be capitalized and italicized. Lagochilascaris minor is not always in italics and lagochilascariasis is randomly capitalized in places.
Line 33-34: You stress that lagochilascariasis is worldwide in the tropics but the review really just focuses on a limited part of South America. Please clear this up in the immediate next sentence.
"lagochilascariasis"
Line 127-128: "Approaches to reduce these changes need to align with local environmental conditions to be efficacious [13-15]."
Are you actually suggesting that climate control of region or the world would be useful in controlling this disease?
Line 139-140: "Barriers to receiving adequate medical care for low socioeconomic groups can worsen the severity of disease and transmission [20,21]."
Explain how "Barriers to receiving adequate medical" make the transmission of the parasite worse?
Line 254-266: This is largely true of all diseases and even the best studied systems. More research will show or not show something and will or will not be helpful. It is also not terribly informative in a review. I guarantee that most research papers say somewhere that "more research is needed" and this is what the paragraph says.
Much of the manuscript is highly speculative. For example Line 231-232 "New human encroachment leading to emergence of Lagochilascariasis hotspots might result new epidemiological patterns [36]."
Sure it might result in new patterns but it might not.
This same type of logic if found throughout. As stated above almost every study will say somewhere that further data or studies "might find important" "might find critical" "Might find important" etc and this is not the value of a literature review because it is just speculation often based on a desire to publish more with additional funding.
Line 254 "vectors such as carnivores" I do not think the word vector is correct here. Carnivores are not directly transmitting these nematodes. they are definitive hosts but not vectors.
Line 364-366: This is an example of where I said make sure what you wrote is saying what you think it says "The public health aspects of Lagochilascariasis are complex and multifactorial, requiring an integrative focus that incorporates environmental factors, management..."
"Public health aspects" then "requiring". The public health aspects do not require anything they are descripting features and random capitalization of Lagochilascariasis.
I stopped at this point because the same issues noted are found throughout. The review is mostly speculative and of low value unless it is grammatically cleaned up and the pure speculation is removed.
Comments on the Quality of English Language
The grammar is curious throughout and there are both broken words with extra spaces, missing words, tense errors, odd word choices, etc. I can not tell if this is the product of AI assistance or poor grammar. I would reject this manuscript as currently written because the grammar and structure of almost every other sentence needs editing and review. As an unpaid reviewer I will not do this work but suggest the editor of the journal or the authors need to carefully read and revise this manuscript line by line to ensure it says what they think it says.
There are repetitive sections to the text in this manuscript to say the same general thing.
The disease "lagochilascariasis" is not a proper noun. It should only be capitalized when starting a sentence.
On the other hand, Lagochilascaris minor is a proper noun, as a species, and would always be capitalized and italicized. Lagochilascaris minor is not always in italics and lagochilascariasis is randomly capitalized in places.
The ENTIRE manuscript needs careful grammatical and typographic review.